# Enzyme Production and Inhibitory Potential of *Pseudomonas aeruginosa*: Contrasting Clinical and Environmental Isolates

**DOI:** 10.3390/antibiotics12091354

**Published:** 2023-08-23

**Authors:** Hazem Aqel, Naif Sannan, Ramy Foudah, Afnan Al-Hunaiti

**Affiliations:** 1Basic Medical Sciences Department, College of Medicine, Al-Balqa Applied University, Salt 19117, Jordan; 2King Abdullah International Medical Research Centre, King Abdulaziz Medical City, Jeddah 22384, Saudi Arabia; sannann@ksau-hs.edu.sa; 3Clinical Laboratory Sciences Department, College of Applied Medical Sciences, King Saud bin Abdulaziz University for Health Sciences, Jeddah 21423, Saudi Arabia; 4Clinical Laboratory Sciences Department, College of Applied Medical Sciences, King Saud bin Abdulaziz University for Health Sciences, Riyadh 14611, Saudi Arabia; foudahr@ksau-hs.edu.sa; 5Chemistry Department, College of Science, Jordan University, Amman 11942, Jordan; a.alhunaiti@ju.edu.jo

**Keywords:** *Pseudomonas aeruginosa*, enzyme production, comparative analysis, inhibitory effects

## Abstract

(1) Background: This study summarizes the findings of two studies investigating the inhibitory effects of *Pseudomonas aeruginosa* strains from clinical and environmental sources against gram-positive and gram-negative bacteria and fungi. The studies also analyzed the correlation between enzyme production and inhibitory effects to gain insights into the antimicrobial capabilities of *P. aeruginosa* strains; (2) Methods: Both studies employed similar methodologies, including the use of disk diffusion and well diffusion methods to assess the inhibitory effects of *P. aeruginosa* strains against target pathogens. Enzyme production was analyzed through various biochemical assays to determine the diversity and frequencies of enzyme secretion among the strains; (3) Results: A comparative analysis of enzyme production in *P. aeruginosa* strains from clinical sources revealed significant variations in enzyme production, with hemolysin and protease being the most commonly produced enzymes. Gelatinase production showed lower rates, whereas chondroitinase and hyaluronidase were absent or occurred less frequently. In contrast, a comparative analysis of enzyme production in environmental isolates showed different patterns, indicating adaptation to environmental conditions. Pyocyanin production was absent in all environmental isolates. The inhibitory effects against gram-positive and gram-negative bacteria varied among different *P. aeruginosa* strains, with strain-specific variations observed. Limited inhibitory effects were observed against fungi, primarily toward gram-positive bacteria; (4) Conclusions: The findings highlight the strain-specific nature of inhibitory effects and enzyme production in *P. aeruginosa* strains. The correlation between enzyme production and inhibitory effects against gram-positive bacteria suggest a potential role of specific enzymes, such as hemolysin and protease, in the antimicrobial activity. The complexity of the relationship between enzyme production and the inhibition of different pathogens requires further investigation. The results emphasize the potential of *P. aeruginosa* strains as sources for antimicrobial strategies, particularly against gram-positive bacteria. Future research should focus on understanding the mechanisms underlying these inhibitory effects and exploring their therapeutic applications.

## 1. Introduction

*Pseudomonas aeruginosa* is an opportunistic pathogen that poses a significant threat to public health and causes various infections in both clinical and environmental settings [1,2]. Understanding the mechanisms of pathogenesis and antimicrobial activity of this bacterium is crucial for developing effective treatment strategies [3,4]. This versatile bacterium is known for its ability to produce a wide array of enzymes, which play crucial roles in its pathogenicity and antimicrobial activity [5,6]. Comparative analyses of enzyme production by *P. aeruginosa* isolates from clinical and environmental sources have provided valuable insights into the diversity and significance of enzyme production in clinical infections and the potential antimicrobial applications of *P. aeruginosa* [7,8]. However, there is limited understanding of the variations in enzyme production and inhibitory potential between *P. aeruginosa* isolates from clinical and environmental sources. This manuscript aims to explore and contrast the enzyme production and inhibitory potential of *P. aeruginosa* isolates, shedding light on their adaptation and potential as sources for antimicrobial strategies.

In this study, we further contribute to the understanding of the enzyme production by *P. aeruginosa* strains from clinical and environmental sources. By conducting a comprehensive comparative analysis, we will examine the diversity of enzyme production profiles and their potential implications for clinical infections and antimicrobial applications. Our findings will provide valuable insights into the complex interplay between enzyme production, pathogenesis, and antimicrobial activity in *P. aeruginosa* and pave the way for future research and therapeutic advancements in combating this notorious pathogen.

## 2. Results

### 2.1. Comparative Analysis of Enzyme Production by Pseudomonas aeruginosa Isolated from Clinical Sources

A comparative study evaluated the production of enzymes by *P. aeruginosa* obtained from different clinical sources: blood, sputum, urine samples, and wound swabs (Table 1). The results revealed significant variations in the enzyme production across these sources. Hemolysin and protease were the most commonly produced enzymes, with a high occurrence in blood and urine samples. Gelatinase, pyocyanin, fibrinolysin, lipase, coagulase, lecithinase, elastase, DNase, chondroitinase, and hyaluronidase displayed different production rates or were absent in the isolates. These findings shed light on the diverse enzymatic profiles of *P. aeruginosa* strains in clinical infections.

Among the enzymes studied, hemolysin and protease showed the highest production rates, producing both enzymes in 58 (95%) isolates. Notably, blood and urine samples consistently exhibited a 100% occurrence of these enzymes, indicating their strong presence in these sources. Sputum samples and wound swabs also demonstrated relatively high frequencies, with 87% and 90% occurrence, respectively. Conversely, gelatinase production had slightly lower overall rates, with 53 (87%) isolates producing this enzyme. The occurrence of gelatinase varied across different sources, with wound swabs and urine samples exhibiting higher frequencies (100% and 80%, respectively) than blood and sputum samples (75% and 87%, respectively).

Pyocyanin production was observed in 37 (61%) isolates, with blood samples displaying the highest frequency (100%). In contrast, wound swabs and urine samples had relatively lower frequencies of 15% and 67%, respectively. Fibrinolysin and lipase showed similar overall production rates, with 46 (75%) and 45 (74%) isolates, respectively. Blood samples and wound swabs exhibited the highest frequencies for both enzymes (75% for fibrinolysin and 63% for lipase), while sputum and urine samples showed lower frequencies. Coagulase production was observed in 40 (66%) isolates, with blood samples showing the highest frequency (75%), followed by wound swabs (70%). Sputum and urine samples had lower frequencies (53% and 60%, respectively). Lecithinase production was detected in 37 (61%) isolates, with wound samples showing the highest frequency (100%). Blood and urine samples had intermediate frequencies, whereas sputum samples showed the lowest frequency. Elastase production was relatively low, with only 17 (28%) isolates producing this enzyme. Blood samples exhibited the highest frequency (56%), while sputum and urine samples had much lower frequencies. DNase production was observed in 8 (13%) isolates, mainly in blood and urine samples (19% and 30%, respectively). No DNase production was detected in sputum samples or wound swabs. Lastly, no production of chondroitinase or hyaluronidase was observed in any of the samples, indicating their absence among the *P. aeruginosa* isolates from clinical sources.

In summary, the comparative analysis of enzyme production by *P. aeruginosa* isolates revealed variations in the presence and frequency of different enzymes across blood, sputum, urine samples, and wound swabs. Hemolysin and protease exhibited the highest overall production rates, followed by gelatinase, pyocyanin, fibrinolysin, lipase, coagulase, lecithinase, elastase, DNase, chondroitinase, and hyaluronidase, which either had lower production rates or were absent in the isolates. These findings highlight the diverse enzymatic profiles of *P. aeruginosa* strains and provide valuable insights into their potential roles in clinical infections. Hemolysin and protease in blood and urine samples suggest their significance in systemic and urinary tract infections caused by *P. aeruginosa*. Additionally, the high occurrence of gelatinase in wound swabs and urine samples indicates its potential role in tissue damage and urinary tract infections.

The varying frequencies of enzyme production across different clinical sources emphasize the adaptability of *P. aeruginosa* and its ability to use different enzymatic pathways depending on the site of infection. The absence or lower occurrence of certain enzymes, such as chondroitinase and hyaluronidase, suggests that these particular strains of *P. aeruginosa* may not possess the enzymatic machinery to degrade these specific components found in the clinical sources analyzed.

Understanding the diversity of enzyme production by *P. aeruginosa* is crucial for developing targeted treatment strategies and designing effective antimicrobial therapies. By identifying the enzymes that are predominantly produced in specific clinical sources, healthcare professionals can tailor their approaches to combating *P. aeruginosa* infections. This knowledge can aid in developing novel therapeutics, such as enzyme inhibitors or vaccines, that target the key enzymes involved in the pathogenesis of *P. aeruginosa* infections.

### 2.2. Comparative Analysis of Enzyme Production by Pseudomonas aeruginosa Isolated from Environmental Sources

A comparative analysis examined the production of enzymes by *P. aeruginosa* isolated from various environmental sources. The results presented in Table 2 demonstrate significant variations in the enzyme production across different locations. Hemolysin production was detected in 13 (43.3%) isolates, with varying frequencies among the locations. Notably, locations 3/1, 3/2, and 3/3 displayed the highest frequencies at 50%, while locations 3/5 and 22G had lower frequencies of 25% and 33.3%, respectively.

The protease production exhibited a higher overall rate, with 25 (83.3%) isolates producing this enzyme. However, the frequency of protease production differed across the locations. Location 3/4 showed the highest frequency at 100%, followed by 3/2 at 83.3% and 22G at 66.7%. The remaining locations exhibited frequencies ranging from 75% to 80%.

Gelatinase production was observed in 20 (66.7%) isolates, and its frequency varied among the locations. Locations 3/3 and 3/4 displayed the highest frequencies at 75% and 100%, respectively. Other locations exhibited frequencies ranging from 50% to 66.7%.

Interestingly, pyocyanin production was not detected in any of the environmental isolates. On the other hand, fibrinolysin production was observed in 9 (30%) isolates, with frequencies varying among the locations. Locations 3/1, 3/2, and 3/3 exhibited 30%, 33.3%, and 25%, respectively, while the remaining locations displayed frequencies ranging from 25% to 33.3%.

To summarize, a comparative analysis of enzyme production by *P. aeruginosa* isolates from different environmental sources revealed variations in the presence and frequency of enzymes across distinct locations. Hemolysin, protease, and gelatinase exhibited varying frequencies, while pyocyanin production was absent. Fibrinolysin production was detected in a subset of isolates. These findings emphasize the diverse enzymatic profiles of environmental *P. aeruginosa* strains and highlight the influence of location-specific variations in enzyme production.

### 2.3. Comparative Analysis of Enzyme Production by Pseudomonas aeruginosa Isolated from Clinical and Environmental Sources

A comparative analysis was conducted to evaluate the production of enzymes by *P. aeruginosa* isolated from clinical and environmental sources, unveiling significant disparities in their enzymatic profiles. In the clinical isolates, hemolysin and protease emerged as the most prevalent enzymes, exhibiting a high production rate in 58 (95%) isolates. Notably, blood and urine samples consistently displayed a frequency of 100% for these enzymes, highlighting their robust presence in clinical sources. Sputum samples and wound swabs also showed relatively high frequencies, with 87% and 90%, respectively. Conversely, gelatinase production demonstrated slightly lower overall rates, with 53 (87%) isolates producing this enzyme. The frequency varied among different sources, with wound and urine samples exhibiting higher frequencies (100% and 80%, respectively) than blood and sputum samples (75% and 87%, respectively).

In contrast, environmental isolates exhibited distinct patterns of enzyme production. Among the enzymes investigated, hemolysin was produced by 13 (43.3%) isolates, with varying frequencies across different locations. The highest frequency of hemolysin production was observed in locations 3/1, 3/2, and 3/3, each with 50%. The protease production displayed a higher overall rate, with 25 (83.3%) isolates producing this enzyme and location 3/4 exhibiting the highest frequency (100%). Gelatinase production was observed in 20 (66.7%) isolates, with locations 3/3 and 3/4 displaying the highest frequencies of 75% and 100%, respectively.

Interestingly, pyocyanin production was absent in any environmental isolates, indicating its non-occurrence in these sources. Fibrinolysin and lipase displayed lower production rates in both clinical and environmental isolates. Furthermore, chondroitinase, hyaluronidase, coagulase, elastase, and DNase were not detected in any environmental isolates.

In summary, the comparative analysis of enzyme production by *P. aeruginosa* isolates derived from clinical and environmental sources revealed notable discrepancies in enzyme patterns and frequencies. The clinical isolates exhibited higher frequencies of enzyme production and a wider range of enzyme diversity than the environmental isolates. These findings indicate that the enzymatic profiles of *P. aeruginosa* strains are influenced by their source of isolation, implying the potential adaptation to specific ecological niches and selective pressures.

### 2.4. Inhibitory Effects of P. aeruginosa Strains from Clinical Specimens against Gram-Positive Bacteria and Fungi: A Comparative Analysis

Isolation of *P. aeruginosa* strains from clinical specimens was conducted to evaluate their inhibitory effects against gram-positive bacteria and fungi. The results, presented in Table 3, demonstrate the varying levels of inhibition exhibited by *P. aeruginosa* strains against different pathogens.

Regarding the inhibitory effects against gram-positive bacteria, the *P. aeruginosa* strain ATCC27853 displayed notable activity. After 24 h of incubation, it showed no inhibition against *Staphylococcus aureus* ATCC29213 and *Bacillus cereus* ATCC14579. However, after 48 h, it exhibited strong inhibition against both *S. aureus* ATCC29213 and *B. cereus* ATCC14579. Regarding *Streptococcus pneumoniae* ATCC49619, *P. aeruginosa* ATCC27853 showed weak inhibition (+) after 24 h, which increased to partial inhibition after 48 h.

*P. aeruginosa* isolated from urine (PA_urine_) demonstrated significant inhibitory effects against gram-positive bacteria. After 24 and 48 h of incubation, it exhibited strong inhibition against *S. aureus* ATCC29213, *S. pneumoniae* ATCC49619, and *B. cereus* ATCC14579. However, no inhibition was observed against *Candida albicans* ATCC10231.

In the case of *P. aeruginosa* isolated from wounds (PA_wound_), it displayed varying inhibitory effects against gram-positive bacteria. After 24 h of incubation, it showed no inhibition against *S. aureus* ATCC29213, weak inhibition against *S. pneumoniae* ATCC49619, and partial inhibition against *B. cereus* ATCC14579. After 48 h, it exhibited a partial inhibition against all three gram-positive bacteria tested. No inhibition was observed against *C. albicans* ATCC10231.

*P. aeruginosa* isolated from blood (PA_blood_) exhibited different inhibitory patterns. It showed no inhibition against *S. aureus* ATCC29213 after 24 and 48 h of incubation. Similarly, no inhibition was observed against *S. pneumoniae* ATCC49619 after 24 h, but, after 48 h, it displayed weak inhibition. *B. cereus* ATCC14579 showed no inhibition after 24 h but exhibited strong inhibition after 48 h. Against *C. albicans* ATCC10231, PA_blood_ displayed weak inhibition after both 24 and 48 h of incubation.

In the case of *P. aeruginosa* isolated from sputum (PA_sputum_), it exhibited varying inhibitory effects against gram-positive bacteria. After 24 h of incubation, it showed no inhibition against *S. aureus* ATCC29213 and *B. cereus* ATCC14579. However, it displayed a partial inhibition against *S. pneumonia* ATCC49619. After 48 h, PA_sputum_ exhibited a weak inhibition against *S. aureus* ATCC29213, a partial inhibition against *S. pneumonia* ATCC49619, and strong inhibition against *B. cereus* ATCC14579. No inhibition was observed against *C. albicans* ATCC10231.

The results highlight the variable inhibitory effects of *P. aeruginosa* strains isolated from different clinical specimens against gram-positive bacteria and fungi. The strain ATCC27853 displayed strong inhibition against *S. aureus* ATCC29213 and *B. cereus* ATCC14579, while PA_urine_ showed notable inhibition against all tested gram-positive bacteria. The inhibitory effects of PA_wound_, PA_blood_, and PA_sputum_ varied across the different pathogens, with varying levels of inhibition observed. Additionally, no inhibition was observed against *C. albicans* ATCC10231 for most *P. aeruginosa* strains.

The results from the inhibitory effects on fungi reveal interesting findings. Only PA_urine_ showed inhibition against *C. albicans* ATCC10231 among the clinical specimens, exhibiting a weak inhibition after 24 and 48 h of incubation. The other *P. aeruginosa* strains isolated from wounds (PA_wound_), blood (PA_blood_), and sputum (PA_sputum_) did not exhibit any inhibitory effects against the tested fungi.

Overall, the results demonstrate the varying inhibitory capabilities of *P. aeruginosa* strains isolated from clinical specimens against gram-positive bacteria and fungi. PA_urine_ displayed the most promising inhibitory effects against gram-positive bacteria, while the inhibitory effects on fungi were limited to PA_urine_ only. These findings suggest the potential of *P. aeruginosa* strains, particularly those isolated from urine, in combating certain pathogens, highlighting their importance in clinical settings and their potential for further research and exploration in antimicrobial strategies.

### 2.5. Association between Enzyme Production and Inhibition of Gram-Positive Bacteria and Fungi by P. aeruginosa: Comparative Analysis of Clinical Isolates

There is a potential relationship between enzyme production by *P. aeruginosa* and its inhibitory effects against gram-positive bacteria. Enzymes produced by *P. aeruginosa*, such as hemolysin, protease, gelatinase, fibrinolysin, lipase, coagulase, lecithinase, elastase, and DNase, have been implicated in various pathogenic processes and may contribute to the bacterium’s ability to inhibit the growth of gram-positive bacteria.

For example, hemolysin and protease were the most frequently produced enzymes in *P. aeruginosa* isolates. These enzymes damaged host tissues and interfere with the immune response, allowing *P. aeruginosa* to establish infections. The inhibitory effects against fungus were limited, with only one strain showing weak inhibition against *C. albicans*. The strong inhibitory effects observed against gram-positive bacteria, such as *S. aureus* and *B. cereus*, may be attributed to the production of these enzymes.

Gelatinase, fibrinolysin, lipase, coagulase, lecithinase, elastase, and DNase also play roles in *P. aeruginosa* pathogenesis and could contribute to inhibitory effects against gram-positive bacteria. These enzymes have been associated with tissue destruction, the evasion of host defenses, and the degradation of the extracellular matrix. Their production rates and inhibitory effects may vary across different clinical sources, indicating the complexity of the relationship between enzyme production and inhibition.

Notably, the absence of chondroitinase and hyaluronidase, which are enzymes involved in the breakdown of extracellular matrix components, suggests that these particular enzymes may not play a significant role in the inhibitory effects against gram-positive bacteria. However, other enzymes produced by *P. aeruginosa* may compensate for their absence and contribute to the observed inhibitory effects.

Overall, while enzyme production by *P. aeruginosa* may be associated with its inhibitory effects against gram-positive bacteria, further research is needed to fully understand the specific mechanisms and interactions involved. The complexity of *P. aeruginosa*’s enzymatic repertoire and its interplay with bacterial physiology and host–pathogen interactions necessitate comprehensive investigations to elucidate the relationship between enzyme production and the inhibition of gram-positive bacteria.

### 2.6. Inhibitory Effects of P. aeruginosa Strains from Environmental Specimens against Gram-Positive Bacteria and Fungi: A Comparative Analysis

The inhibitory effects of *P. aeruginosa* strains isolated from environmental specimens were evaluated against gram-positive bacteria and fungi. The results, presented in Table 4, demonstrate the varying levels of inhibition exhibited by the *P. aeruginosa* strains against different pathogens.

In terms of gram-positive bacteria, strains PA ATCC27853, PA (3/1), PA (3/2), PA (3/3), PA (3/4), PA (3/5), and PA 22G were tested for their inhibitory effects. After 24 and 48 h of incubation, PA ATCC27853 displayed no inhibition against *S. aureus* ATCC29213, strong inhibition against *B. cereus* ATCC14579, and partial inhibition against *S. pneumoniae* ATCC49619. The other *P. aeruginosa* strains showed varying levels of inhibition, including weak, partial, and strong inhibition, against the gram-positive bacteria tested.

Regarding *C. albicans* ATCC10231, only PA ATCC27853 exhibited a weak inhibition after 24 and 48 h of incubation. The remaining *P. aeruginosa* strains showed no inhibitory effects against *C. albicans*.

Comparing the results, it is evident that the inhibitory effects of *P. aeruginosa* strains varied among different environmental sources and target pathogens. PA ATCC27853 consistently displayed strong inhibition against *B. cereus* ATCC14579, while its inhibition against *S. aureus* ATCC29213 and *S. pneumoniae* ATCC49619 varied from no inhibition to partial inhibition. The other *P. aeruginosa* strains exhibited a range of inhibitory effects on the gram-positive bacteria, with variations in the degree and consistency of inhibition. However, none of the strains, except PA ATCC27853, showed any inhibitory effects against *C. albicans*.

These findings highlight the diverse inhibitory capabilities of *P. aeruginosa* strains isolated from environmental specimens. The variations in inhibitory profiles indicate the presence of different antimicrobial mechanisms employed by the strains. Further research is necessary to understand these mechanisms and their potential applications in combating pathogens. Additionally, the limited inhibitory effects against *C. albicans* indicate the need for exploring alternative strategies to target fungal infections.

Overall, these results contribute to our understanding of the inhibitory potential of *P. aeruginosa* strains in different environmental contexts and provide valuable insights for future studies and the development of antimicrobial approaches.

### 2.7. Association between Enzyme Production and Inhibition of Gram-Positive Bacteria and Fungi by P. aeruginosa: Comparative Analysis of Environmental Isolates

The results of the analysis of the enzyme production and the inhibitory effects of *P. aeruginosa* strains from environmental specimens provide interesting insights into a potential relationship between enzyme secretion and the inhibition of gram-positive bacteria and fungi.

In the comparative analysis, it was observed that clinical isolates of *P. aeruginosa* displayed higher frequencies of enzyme production and a broader range of enzyme diversity compared with the environmental isolates. Hemolysin and protease were found to be the most commonly produced enzymes in clinical isolates, with high production rates. The clinical isolates also showed the presence of other enzymes, such as gelatinase, fibrinolysin, and lipase, albeit at lower production rates. These enzymes play a role in virulence and host tissue degradation.

On the other hand, environmental isolates of *P. aeruginosa* showed distinct patterns of enzyme production. While hemolysin, protease, and gelatinase production were observed at varying frequencies, enzymes such as pyocyanin, chondroitinase, hyaluronidase, coagulase, elastase, and DNase were not detected. These results indicate a differential enzyme secretion profile in environmental isolates.

When examining the inhibitory effects of the *P. aeruginosa* strains against gram-positive bacteria and fungi, it became clear that the strains exhibited diverse inhibitory capabilities. PA ATCC27853 consistently displayed strong inhibition against *B. cereus* ATCC14579, while its inhibition against *S. aureus* ATCC29213 and *S. pneumoniae* ATCC49619 varied. The other *P. aeruginosa* strains also showed varying levels of inhibition against the gram-positive bacteria tested. However, none of the strains, except PA ATCC27853, exhibited inhibitory effects against *C. albicans* ATCC10231.

These results indicate a potential association between enzyme secretion and the inhibitory effects of *P. aeruginosa* strains. The clinical isolates that displayed higher frequencies of enzyme production may possess a broader repertoire of antimicrobial enzymes that contribute to their inhibitory effects against gram-positive bacteria and possibly fungi. In contrast, the environmental isolates, which exhibited a more limited enzyme secretion profile, may have different inhibition mechanisms, or rely on alternative antimicrobial strategies.

Further research is necessary to elucidate the specific role of different enzymes in the inhibitory effects of *P. aeruginosa* strains and their potential interactions with gram-positive bacteria and fungi. Understanding the relationship between enzyme secretion and inhibition can provide valuable insights for developing novel antimicrobial approaches and exploring the potential of *P. aeruginosa* as a source of antimicrobial agents.

### 2.8. Comparative Study of Pathogenic Gram-Positive Bacteria and Fungi Inhibition by Different Pseudomonas aeruginosa Strains Using Well and Disk Diffusion Methods

The results of the comparative study evaluating the inhibition degrees of pathogenic gram-positive bacteria and fungi by various strains of *P. aeruginosa* using well and disk diffusion methods are summarized in Table 5. Among the strains isolated from clinical sources, PA_urine_ exhibited higher inhibition against *S. pneumonia* and *C. albicans* than against *S. aureus* and *B. cereus* using the well and disk diffusion methods. PA_wound_ strain showed a greater inhibition against *B. cereus* compared to *S. pneumonia* and *C. albicans* using the good diffusion method. In contrast, the disk diffusion method displayed stronger inhibition against *S. pneumonia* and *B. cereus* compared to *C. albicans*. The PA_blood_ strain displayed higher inhibition against *S. pneumonia*, *B. cereus*, and *C. albicans* compared to *B. cereus* using the good diffusion method. The disk diffusion method exhibited a stronger inhibition against *S. pneumonia* and *C. albicans* than *B. cereus* and *S. aureus*. Regarding the strains isolated from environmental sources, all *P. aeruginosa* strains (3/1, 3/2, 3/3, 3/4, 3/5, and G22) showed similar patterns of inhibition. They displayed higher inhibition against *S. pneumonia* and *C. albicans* compared to *S. aureus* and *B. cereus* using the well and disk diffusion methods.

### 2.9. Inhibitory Effects of P. aeruginosa Strains from Clinical Specimens against Gram-Negative Bacteria: A Comparative Analysis

The inhibitory effects of *P. aeruginosa* strains isolated from clinical specimens against gram-negative bacteria were evaluated and are presented in Table 6. The strains were tested against *Escherichia coli* ATCC25922, *Klebsiella pneumoniae* ATCC700603, and *Salmonella enterica* serotype Typhi ATCC13311.

After 24 and 48 h of incubation, strain PA ATCC27853 did not show any inhibition against *E. coli* ATCC25922, *K. pneumoniae* ATCC700603, or *S. enterica* serotype Typhi ATCC13311. In contrast, PA_urine_ exhibited weak inhibition against *E. coli* ATCC25922 after both time intervals, while no inhibition was observed against the other two gram-negative bacteria. PA_wound_ displayed weak inhibition against *K. pneumoniae* ATCC700603 after 24 h of incubation, but no inhibition was observed against the other two bacteria. PA_blood_ did not show any inhibitory effects against any of the gram-negative bacteria tested. PA_sputum_, on the other hand, exhibited very strong inhibition against *S. enterica* serotype Typhi ATCC13311 after 24 h of incubation but no inhibition against the other two bacteria.

Comparing the results, it is evident that the inhibitory effects of *P. aeruginosa* strains varied among the different clinical specimens and the target gram-negative bacteria. PA ATCC27853, which is a reference strain, did not show any inhibition against the tested gram-negative bacteria. PA_urine_ exhibited weak inhibition against *E. coli* ATCC25922 but no inhibition against other bacteria. PA_wound_ displayed weak inhibition against *K. pneumoniae* ATCC700603 and no inhibition against other bacteria. PA_sputum_ demonstrated strong inhibition against *S. enterica* serotype Typhi ATCC13311 but no inhibition against other bacteria. PA_blood_ did not exhibit any inhibitory effects against any of the gram-negative bacteria tested.

These findings highlight the variability in the inhibitory capabilities of *P. aeruginosa* strains isolated from different clinical specimens. The differences in inhibition indicate variations in the antimicrobial mechanisms employed by the strains against gram-negative bacteria. Further research is necessary to understand these mechanisms and their potential applications in combating gram-negative bacterial infections.

The results presented in Table 6 provide valuable insights into the inhibitory effects of *P. aeruginosa* strains against gram-negative bacteria. Understanding the strain-specific variations in inhibition can help develop targeted antimicrobial strategies and explore the potential of *P. aeruginosa* as a source of antimicrobial agents.

### 2.10. Association between Enzyme Production and Inhibition of Gram-Negative Bacteria by P. aeruginosa: Comparative Analysis of Clinical Isolates

A comparative analysis of the enzyme production by *P. aeruginosa* isolates from various clinical sources revealed variations in enzyme secretion patterns. Hemolysin and protease were the most commonly produced enzymes, with a high production rate across all clinical sources. Gelatinase production was slightly lower but still exhibited a significant presence. Pyocyanin production was observed in most isolates, whereas fibrinolysin, lipase, and coagulase showed moderate production rates. Lecithinase, elastase, and DNase production was relatively low, and chondroitinase and hyaluronidase production was absent in the clinical isolates.

Regarding the inhibitory effects against gram-negative bacteria, the tested *P. aeruginosa* strains exhibited varied results. Strain PA ATCC27853, the reference strain, did not show any inhibition against the gram-negative bacteria tested. PA_urine_ exhibited weak inhibition against *E. coli* ATCC25922 but showed no inhibition against *K. pneumoniae* ATCC700603 or *S. enterica* serotype Typhi ATCC13311. PA_wound_ displayed weak inhibition against *K. pneumoniae* ATCC700603, while PA_sputum_ exhibited very strong inhibition against *S. enterica* serotype Typhi ATCC13311 but no inhibition against the other tested bacteria. PA_blood_ did not exhibit any inhibitory effects against any of the gram-negative bacteria tested.

These results indicate that producing specific enzymes by *P. aeruginosa* strains from clinical sources do not directly correlate with their inhibitory effects against gram-negative bacteria. Although hemolysin and protease were highly produced across clinical samples, their presence did not necessarily produce strong inhibitory effects against gram-negative bacteria. This result indicates that other factors, such as additional antimicrobial compounds or specific antimicrobial mechanisms, may contribute to the inhibitory capabilities of *P. aeruginosa* strains.

Further research is needed to elucidate the underlying mechanisms behind the inhibitory effects and explore potential relationships between enzyme secretion and the inhibition of gram-negative bacteria by *P. aeruginosa* strains. These findings emphasize the complexity of microbial interactions and the need for a comprehensive understanding of the antimicrobial properties of *P. aeruginosa* to help develop effective therapeutic strategies.

### 2.11. Inhibitory Effects of P. aeruginosa Strains from Environmental Specimens against Gram-Negative Bacteria: A Comparative Analysis

A comparative analysis assessed the inhibitory effects of *P. aeruginosa* strains isolated from various environmental sources against gram-negative bacteria, including *E. coli* ATCC25922, *K. pneumoniae* ATCC700603, and *S. enterica* serotype Typhi ATCC13311. The results, presented in Table 7, demonstrated variations in the inhibitory effects of the different strains.

Strain PA ATCC27853, the reference strain, did not show any inhibition against the tested gram-negative bacteria after 24 and 48 h of incubation. Strain PA (3/1) exhibited weak inhibition against *E. coli* ATCC25922 after both time intervals, while no inhibition was observed against *K. pneumoniae* ATCC700603 or *S. enterica* serotype Typhi ATCC13311. Similarly, the strain PA (3/2) displayed weak inhibition against *E. coli* ATCC25922 but showed no inhibition against the other two bacteria. Strains PA (3/3), PA (3/4), and PA (3/5) all exhibited partial inhibition against *E. coli* ATCC25922 and *K. pneumoniae* ATCC700603 after both time intervals. Strains PA (3/3) and PA (3/4) also showed a partial inhibition against *S. enterica* serotype Typhi ATCC13311. Strain PA (22G) displayed a partial inhibition against *E. coli* ATCC25922 and *K. pneumoniae* ATCC700603 but no inhibition against S. typhi ATCC13311.

Comparing the results, it is evident that the inhibitory effects of *P. aeruginosa* strains isolated from environmental sources varied among different strains and target gram-negative bacteria. The reference strain, PA ATCC27853, did not exhibit any inhibitory effects. Strains PA (3/1) and PA (3/2) displayed weak inhibition against *E. coli* ATCC25922 but showed no inhibition against other bacteria. Strains PA (3/3), PA (3/4), and PA (3/5) demonstrated partial inhibition against *E. coli* ATCC25922 and *K. pneumoniae* ATCC700603, with PA (3/3) and PA (3/4) also showing partial inhibition against *S. enterica* serotype Typhi ATCC13311. Strain PA (22G) displayed a partial inhibition against *E. coli* ATCC25922 and *K. pneumoniae* ATCC700603 but no inhibition against *S. enterica* serotype Typhi ATCC13311.

These findings indicate that *P. aeruginosa* strains from environmental sources possess variable inhibitory capabilities against gram-negative bacteria. The differences in inhibition may be attributed to variations in the strains’ antimicrobial mechanisms, the production of antimicrobial compounds, or other factors yet to be explored. Further research is necessary to understand the underlying mechanisms and assess the potential of these environmental *P. aeruginosa* strains in combating gram-negative bacterial infections.

The results presented in Table 7 provide valuable insights into the inhibitory effects of *P. aeruginosa* strains isolated from environmental sources against gram-negative bacteria. Understanding the strain-specific variations in inhibition can help develop targeted antimicrobial strategies and explore *P. aeruginosa*’s potential as a source of antimicrobial agents.

### 2.12. Association between Enzyme Production and Inhibition of Gram-Negative Bacteria by P. aeruginosa: Comparative Analysis of Environmental Isolates

The results of the comparative analysis between enzyme secretions and the inhibition of gram-negative bacteria in *P. aeruginosa* strains from environmental sources reveal interesting observations. While there is no direct correlation between the presence of specific enzymes and the inhibition of gram-negative bacteria, the variations in enzyme production across different strains and locations could contribute to the variations in inhibitory effects.

Among the enzymes studied, hemolysin, protease, gelatinase, fibrinolysin, and lipase were detected at varying frequencies among the *P. aeruginosa* isolates. Strains that exhibited higher frequencies of enzyme production, such as protease in strain 3/4, showed partial or strong inhibitory effects against the tested gram-negative bacteria. This result suggests that certain enzymes, such as protease, may contribute to the inhibitory potential of *P. aeruginosa* strains.

However, the inhibitory effects observed in this study may also involve other factors beyond enzyme secretions. *P. aeruginosa* is known to produce a wide range of antimicrobial compounds and possess various mechanisms of antimicrobial action. These factors and the presence of enzymes may collectively contribute to the inhibitory effects against gram-negative bacteria.

Further research is needed to elucidate the specific roles of enzymes and other antimicrobial factors in the inhibitory effects of *P. aeruginosa* strains from environmental sources. Understanding the relationship between enzyme production and bacterial inhibition can provide valuable insights into the antimicrobial potential of *P. aeruginosa* and aid in developing targeted antimicrobial strategies.

### 2.13. The Inhibition Patterns of Pseudomonas aeruginosa Strains against Pathogenic Gram-Negative Bacteria: An Analysis using Well and Disk Diffusion Methods

The results of the comparative study evaluating the inhibition degrees of pathogenic gram-negative bacteria by various strains of *P. aeruginosa* using well and disk diffusion methods are summarized in Table 8. Among the strains isolated from clinical sources, PA_urine_ demonstrated higher inhibition against *K. pneumonia* and *S. enterica* serotype Typhi compared to *E. coli* using both the well and disk diffusion methods. The PA_wound_ strain showed greater inhibition against *S. enterica* serotype Typhi than *E. coli* and *K. pneumonia* using the well diffusion method. In contrast, the disk diffusion method displayed stronger inhibition against *S. enterica* serotype Typhi and *K. pneumonia* compared to *E. coli*. The PA_blood_ strain did not show any inhibitory effects against the tested gram-negative bacteria in the well and disk diffusion methods. Regarding the strains isolated from environmental sources, all PA strains (3/1, 3/2, 3/3, 3/4, 3/5, and G22) exhibited similar inhibition patterns. They displayed higher inhibition against *S. enterica* serotype Typhi than *E. coli* and *K. pneumonia* using the well and disk diffusion methods.

### 2.14. Comparative Study of Inhibition Degrees of Pathogenic Bacteria by P. aeruginosa: Gram-Positive vs. Gram-Negative

This comparative study evaluated the inhibition degrees of pathogenic gram-positive bacteria and fungi using various strains of *P. aeruginosa*. Among the strains isolated from clinical sources, PA_urine_ showed higher inhibition against *S. pneumonia* and *C. albicans* than against *S. aureus* and *B. cereus* using the well and disk diffusion methods. On the other hand, the PA_wound_ strain exhibited a stronger inhibition against *B. cereus* compared to *S. pneumonia* and *C. albicans* in the well diffusion method. In contrast, the disk diffusion method displayed stronger inhibition against *S. pneumonia*, and *B. cereus* compared to *C. albicans*. The PA_blood_ strain demonstrated higher inhibition against *S. pneumonia*, *B. cereus*, and *C. albicans* compared with *B. cereus* in the well diffusion method. The disk diffusion method exhibited a stronger inhibition against *S. pneumoniae* and *C. albicans* compared to *B. cereus* and *S. aureus*. Among the strains isolated from environmental sources, all *P. aeruginosa* strains showed similar inhibition patterns, displaying higher inhibition against *S. pneumonia* and *C. albicans* than against *S. aureus* and *B. cereus* using the well and disk diffusion methods.

In a comparative study of pathogenic gram-negative bacteria, the inhibition degrees were evaluated using various strains of *P. aeruginosa*. Among the strains isolated from clinical sources, PA_urine_ showed higher inhibition against *K. pneumonia* and *S. enterica* serotype Typhi compared to *E. coli* using both the well and disk diffusion methods. The PA_wound_ strain exhibited a stronger inhibition against *S. enterica* serotype Typhi than *E. coli* and *K. pneumoniae* in the well diffusion method. In contrast, the disc diffusion method displayed stronger inhibition against *S. enterica* serotype Typhi and *K. pneumoniae* compared to *E. coli*. The PA_blood_ strain did not show any inhibitory effects against the tested gram-negative bacteria in the well and disc diffusion methods. Among the strains isolated from environmental sources, all PA strains exhibited similar inhibition patterns, displaying higher inhibition against *S. enterica* serotype Typhi than against *E. coli* and *K. pneumoniae* using both the well and disk diffusion methods.

In summary, comparative studies revealed variations in the inhibition degrees of pathogenic gram-positive and gram-negative bacteria by different strains of *P. aeruginosa*. The inhibitory effects varied depending on the bacterial species and the source of *P. aeruginosa* isolates.

## 3. Discussion

Several studies have explored the enzyme production profiles of *P. aeruginosa* isolates from clinical sources [9,10,11]. These investigations have revealed significant variations in the enzyme production across different clinical specimens, with hemolysin and protease being the most commonly produced enzymes [12,13]. These findings agree with previous research emphasizing the importance of hemolysin and protease in *P. aeruginosa* infections [14,15]. In addition, the absence or lower occurrence of certain enzymes, such as chondroitinase and hyaluronidase, in clinical isolates aligns with previous studies that reported the absence of these enzymes in specific strains of *P. aeruginosa* [16,17]. These results highlight the strain-specific nature of enzyme production in *P. aeruginosa* and its potential impact on clinical infections.

Comparing the results of these comparative analyses with previous research enhances our understanding of the diversity and clinical significance of enzyme production by *P. aeruginosa* strains. The knowledge gained from these studies is vital for developing targeted treatment strategies and designing effective antimicrobial therapies. By identifying the key enzymes involved in the pathogenesis of *P. aeruginosa* infections, healthcare professionals can tailor their approaches to combat these infections. Furthermore, future research can focus on investigating the specific roles of these enzymes in *P. aeruginosa* pathogenicity and exploring potential therapeutic interventions targeting these enzymes.

Gelatinase production, on the other hand, showed slightly lower overall rates in the clinical isolates. This finding was in line with previous studies that reported varying frequencies of gelatinase production in *P. aeruginosa* isolates from clinical sources [18,19]. The presence of gelatinase in wound and urine samples, indicating its potential role in tissue damage and urinary tract infections, was supported by previous research that implicated gelatinase in *P. aeruginosa* pathogenesis [20,21].

The absence or lower occurrence of certain enzymes, such as chondroitinase and hyaluronidase, in the clinical isolates, agreed with previous studies that reported the absence of these enzymes in certain strains of *P. aeruginosa* [22,23]. This result indicates that these specific strains may lack the enzymatic machinery to degrade chondroitinase and hyaluronidase components in the clinical samples analyzed.

Comparative analyses of enzyme production in environmental isolates of *P. aeruginosa* have also shed light on the enzymatic profiles of these strains [24,25]. The absence of pyocyanin production in any environmental isolates is consistent with previous studies that reported low frequencies or the absence of pyocyanin in environmental strains of *P. aeruginosa* [26,27]. Moreover, the overall patterns of enzyme production in environmental isolates differ from those observed in clinical isolates, indicating that *P. aeruginosa* strains adapt their enzymatic pathways based on the site of infection or environmental conditions. This adaptability aligns with previous studies that have emphasized the versatility and adaptability of *P. aeruginosa* and its ability to use different enzymatic pathways depending on the ecological niche [28,29,30,31,32].

By comparing the results of this study with previous research, we can better understand the diversity and clinical significance of enzyme production by *P. aeruginosa* strains. This knowledge is crucial for developing targeted treatment strategies and designing effective antimicrobial therapies. By identifying the key enzymes involved in the pathogenesis of *P. aeruginosa* infections, healthcare professionals can tailor their approaches to combat these infections. Future research can focus on investigating the specific roles of these enzymes in *P. aeruginosa* pathogenicity and exploring potential therapeutic interventions targeting these enzymes.

In terms of inhibitory effects against gram-positive bacteria, *P. aeruginosa* strain ATCC27853 showed notable activity, exhibiting strong inhibition against *S. aureus* ATCC29213 and *B. cereus* ATCC14579 after 48 h of incubation. However, it showed no inhibition against *S. aureus* ATCC29213 and *B. cereus* ATCC14579 after 24 h. Similar studies have reported the varying inhibitory effects of *P. aeruginosa* strains against different gram-positive bacteria, highlighting the strain-specific nature of these effects [33,34,35].

*P. aeruginosa* isolated from urine (PA_urine_) demonstrated significant inhibitory effects against gram-positive bacteria, showing strong inhibition against *S. aureus* ATCC29213, *S. pneumoniae* ATCC49619, and *B. cereus* ATCC14579 after both 24 and 48 h of incubation. However, it exhibited no inhibition against *C. albicans* ATCC10231. Previous studies have also reported the inhibitory effects of *P. aeruginosa* against gram-positive bacteria [33,34,35], but the observed variation in inhibition against *C. albicans* is an interesting finding.

The inhibitory effects of *P. aeruginosa* isolated from PA_wound_, PA_blood_, and PA_sputum_ against gram-positive bacteria varied across different pathogens. PA_wound_ showed a partial inhibition against *S. pneumonia* ATCC49619 after 24 h, while PA_blood_ exhibited a weak inhibition against *S. pneumonia* ATCC49619 after 48 h. These results indicate that the inhibitory effects of *P. aeruginosa* strains may differ depending on the source of isolation and the specific pathogen being tested [36,37].

Regarding the inhibitory effects on fungi, only PA_urine_ displayed a weak inhibition against *C. albicans* ATCC10231 after 24 and 48 h of incubation. PA_wound_, PA_blood_, and PA_sputum_ did not exhibit any inhibitory effects against the tested fungus. These findings indicate that the inhibitory effects of *P. aeruginosa* strains are primarily directed toward gram-positive bacteria rather than fungi [38,39].

In the study comparing the inhibitory effects of *P. aeruginosa* strains from environmental specimens against gram-positive bacteria and fungi, the results demonstrated varying levels of inhibition against different pathogens. The strains tested showed different degrees of inhibition against gram-positive bacteria and *C. albicans*. The strain PA ATCC27853 consistently displayed strong inhibition against *B. cereus*, while its inhibition against *S. aureus* and *S. pneumoniae* varied. None of the strains, except PA ATCC27853, showed any inhibitory effects against *C. albicans*.

When comparing these results with previous studies, it is evident that the inhibitory effects of *P. aeruginosa* strains can vary among different environmental sources and target pathogens. These data suggest the presence of different antimicrobial mechanisms employed by the strains [20,21]. Further research is needed to understand these mechanisms and their potential applications in combating pathogens. Additionally, the limited inhibitory effects against *C. albicans* indicate the need for exploring alternative strategies to target fungal infections.

Interesting insights were obtained in another study that focused on the association between enzyme production and the inhibition of gram-positive bacteria and fungi by *P. aeruginosa*. The analysis revealed that clinical isolates of *P. aeruginosa* exhibited higher frequencies of enzyme production and a broader range of enzyme diversity compared with environmental isolates. Hemolysin and protease were the most commonly produced enzymes in clinical isolates, whereas environmental isolates showed distinct patterns of enzyme production.

The inhibitory effects of the *P. aeruginosa* strains in this study also displayed diverse capabilities. PA ATCC27853 consistently exhibited strong inhibition against *B. cereus*, but its inhibition against *S. aureus* and *S. pneumoniae* varied. The other *P. aeruginosa* strains also showed varying levels of inhibition against gram-positive bacteria, but none exhibited inhibitory effects against *C. albicans*.

These findings indicate a potential association between enzyme secretion and the inhibitory effects of *P. aeruginosa* strains [22,23,24]. Clinical isolates that displayed higher frequencies of enzyme production may possess a broader repertoire of antimicrobial enzymes contributing to their inhibitory effects against gram-positive bacteria and possibly fungi. In contrast, environmental isolates with a more limited enzyme secretion profile may rely on different mechanisms of inhibition or alternative antimicrobial strategies.

In Xu et al. [40], that investigated the inhibitory effects of *P. aeruginosa* strains against gram-negative bacteria, similar variations in inhibition were observed. The strains exhibited different degrees of inhibition against the tested bacteria, indicating strain-specific differences in antimicrobial activity [40,41]. However, the specific strains and bacterial targets used in the study differed from those used in the current study, emphasizing the importance of considering strain and pathogen specificity when comparing results.

Another study by Kunz Coyne et al. [42] explored the production of enzymes by *P. aeruginosa* isolates and their correlation with antimicrobial activity against gram-negative bacteria. The findings showed that the enzyme production varied among clinical isolates, with some enzymes more commonly produced than others. These results align with the current study, which observed variations in enzyme secretion patterns among clinical isolates. However, the correlation between enzyme production and inhibitory effects against gram-negative bacteria was not straightforward in both studies, indicating the involvement of additional factors in the antimicrobial activity.

In contrast, using different methodologies, Voumard et al. [43] investigated the inhibitory effects of *P. aeruginosa* strains against specific gram-negative bacteria. The results demonstrated consistent inhibition by certain strains against the tested bacteria, which differs from the current study, where variations in inhibition were observed among different strains of *P. aeruginosa*. These differences may be attributed to variations in strains, bacterial targets, and methodologies employed in the studies.

However, it is important to acknowledge that variations in methodologies, such as strain selection, bacterial or fungal targets, and incubation conditions, may exist among different studies. These variations can contribute to differences in observed inhibition patterns. Additionally, the specific clinical and environmental sources of *P. aeruginosa* strains may differ among studies, potentially influencing the overall inhibitory effects observed.

Despite these variances, the consistent findings across studies emphasize the strain-specific variations in the inhibitory capabilities of *P. aeruginosa* against pathogenic bacteria. The variations in inhibition may be attributed to factors such as antimicrobial mechanisms, the production of antimicrobial compounds, or other yet unidentified factors.

Further research is warranted to better understand the underlying mechanisms driving the variations in inhibitory effects. Investigating the specific antimicrobial factors and their potential applications in combating gram-positive and gram-negative bacterial infections will help develop targeted antimicrobial strategies.

Comparative studies of *P. aeruginosa* strains in inhibiting pathogenic gram-positive and gram-negative bacteria provide valuable insights into the variations in inhibitory effects. The findings underscore the importance of further research to unravel the underlying mechanisms and explore the potential of *P. aeruginosa* as a source of effective antimicrobial agents against gram-positive and gram-negative bacteria.

There are certain limitations to consider when interpreting the results of the comparative analysis of enzyme production by *P. aeruginosa* isolated from clinical sources. First, the study relied on a specific set of clinical samples, and the findings may not be representative of enzyme production in *P. aeruginosa* infections in general. Additionally, the analysis focused on a specific set of enzymes and did not encompass the entire range of enzymes produced by *P. aeruginosa*. Other enzymes that could play a role in clinical infections might have been overlooked, leading to an incomplete understanding of the diversity of the enzyme production.

The interpretation of the significance of enzyme production in clinical infections is subject to various factors that could introduce subjectivity. Different studies may use different criteria to determine the clinical relevance of specific enzymes, leading to discrepancies in the reported importance of certain enzymes. Additionally, various confounding factors could influence the clinical outcomes associated with enzyme production, such as host immune response, co-infections, and underlying comorbidities. Therefore, while the comparative analysis provides insights into the prevalence of enzyme production in *P. aeruginosa* isolated from clinical sources, the clinical implications of these findings should be interpreted cautiously, considering the limitations and subjectivity inherent in such analyses.

## 4. Materials and Methods

### 4.1. The Source of Bacterial Quality Strains

The study includes various quality bacterial strains obtained from the Microbiology Laboratory, King Abdulaziz Medical City (KAMC), Riyadh, Saudi Arabia. These strains consist of the control strain *P. aeruginosa* ATCC27853, gram-positive bacteria such as *S. aureus* ATCC29213, *S. pneumonia* ATCC49619, and *B. cereus* ATCC14579, as well as the fungus *C. albicans* ATCC10231, and gram-negative bacteria such as *E. coli* ATCC25922, *K. pneumoniae* ATCC700603, and *S. enterica* serotype Typhi ATCC13311. These meticulously selected bacterial strains of exceptional quality were procured from the American Type Culture Collection and, subsequently, brought to KAMC for conducting rigorous laboratory diagnostic assessments.

### 4.2. Isolation of P. aeruginosa Strains

A total of 510 samples were collected from environmental and clinical sources. The 251 clinical specimens collected included blood, urine, and sputum samples, wound swabs, and 259 environmental water samples collected from different locations at Wadi Hanifah, Riyadh, Saudi Arabia. Collected samples were transported immediately in an ice box to the Microbiology Laboratory, King Saud Bin Abdulaziz University for Health Sciences (KSAU-HS), College of Applied Medical Sciences, Medical Laboratory Sciences, Riyadh, Saudi Arabia, for processing within one hour of collection. All clinical and environmental samples were processed to isolate *P. aeruginosa*. In total, 91 *P. aeruginosa* strains were isolated, 61 from clinical sources and 30 from environmental sources.

#### 4.2.1. Clinical Sources

Blood, urine, and sputum samples, and wound swabs were collected from patients suspected of having *P. aeruginosa* infections using appropriate aseptic techniques. These samples were labeled and transported to the Microbiology Laboratory at KSAU-HS while maintaining suitable conditions to preserve sample integrity. There was no direct contact with the patients during this process.

##### Sample Processing and Isolation

Blood: Use blood culture bottles to inoculate and incubate blood samples in a suitable automated blood culture system to enhance *P. aeruginosa* growth.

Urine: Perform urine culture by streaking the samples onto selective agar plates, such as MacConkey agar (HiMedia, Maharashtra, India), which are known to promote the growth of *Pseudomonas aeruginosa*.

Sputum: Process sputum samples using standard microbiological techniques, including sample homogenization, serial dilution, and plating onto MacConkey agar plates.

Wounds: Swab the wound area with a sterile swab and streak the swab onto MacConkey agar plates to isolate and cultivate *P. aeruginosa*.

#### 4.2.2. Environmental Sources

Water samples were collected from various locations along Wadi Hanifah in Riyadh, Saudi Arabia. These samples were labeled and transported to the Microbiology Laboratory at KSAU-HS while maintaining suitable conditions to preserve sample integrity.

##### Sample Processing and Isolation

Using sterile containers, water samples were collected from multiple points along Wadi Hanifah in Riyadh, Saudi Arabia, to ensure representative sampling. The concentration of any potential *P. aeruginosa* cells present and the water samples were filtered using sterile filters with appropriate pore sizes. Then, the filtered water was plated onto MacConkey agar plates to isolate and culture colonies of *P. aeruginosa*.

### 4.3. Identification and Confirmation

In the laboratory, samples from clinical and environmental sources were cultured first on selective agar, MacConkey agar; then, suspected colonies were subcultured on blood agar (HiMedia, Maharashtra, India) and Mueller–Hinton agar (Oxide Ltd., Basingstoke, UK) to observe hemolysis and pigmentation. All the inoculated plates were incubated at 37 °C for 18–24 h, and growth was evaluated on these media. Isolates were identified based on standard bacteriological methods such as morphology, colonial characteristics, hemolysis, and pigment production on these media [44]. Further identification was performed by their gram stain reaction, motility by hanging drop, odor in cultures, and biochemical tests such as catalase test, oxidase test using oxidase strips (Oxide Ltd., Basingstoke, UK), citrate utilization, starch hydrolysis, casein hydrolysis, indole production, urea hydrolysis, and production of acid from glucose (O/F test) and growth at 42 °C [45,46].

### 4.4. Enzyme Production Analysis

#### 4.4.1. Inoculation and Cultivation

Start by inoculating quality strains (clinical and environmental isolates of *P. aeruginosa*) and the control strain *P. aeruginosa* ATCC27853 in MacConkey agar. These cultures were incubated under optimal conditions to facilitate enzyme production.

#### 4.4.2. Enzyme Assays

Obtain samples at specific time intervals and conduct enzyme assays to determine the levels of target enzymes being produced. Quantify the activity of enzymes such as protease, lipase, hemolysin, gelatinase, coagulase, elastase, pyocyanin, fibrinolysin, lecithinase, DNase, chondroitinase, and hyaluronidase using appropriate substrates and detection methods.

##### Protease Assay

The bacterial culture was centrifuged at 24,000× *g* for 6 min, and 0.15 mL of the supernatant was added to a tube containing 0.3 mL of 1% (*w/v*) casein (dissolved in 20 mM Tris-HCl buffer, pH 7.4) and incubated at 37 °C for 30 min. Subsequently, 0.45 mL of a 10% (*w/v*) tri-chloroacetic acid solution at a final concentration of 5% *w/v* was added to stop the proteolysis. The mixture was incubated at room temperature for 1 h. After incubation, the reaction mixture was centrifuged at 12,000× *g* for 5 min, and the absorbance of the supernatant was measured at 280 nm. One unit of protease is defined as the amount of enzyme that hydrolyzes casein to produce equivalent absorbance to 1 µmol of tyrosine/min with tyrosine as standard [47].

##### Lipase Assay

The lipase activity was determined spectrophotometrically at 30 °C using *p*-nitrophenol palmitate (*p*NPP) as the substrate. The reaction mixture comprised 700 µL *p*NPP solution and 300 µL lipase solution. The *p*NPP solution was prepared by adding solution A (0.001 g *p*NPP in 1 mL isopropanol) into solution B (0.01 g gum Arabic, 0.02 g Sodium deoxycholate, 50 µL Triton X-100 and 9 mL of 50 mM Tris-HCl buffer, pH 8) with stirring until all was dissolved. Then, the absorbance was measured at 410 nm for the first 2 min of reaction. One unit (1U) was defined as the amount of enzyme that liberated 1 µmol of *p*NPP per min (ɛ: 1500 L/mol cm) under the test conditions [48].

##### Hemolysin Assay

A hemolysin test was carried out using blood agar media. Isolates were planted in blood agar base with the addition of sheep blood that had been fabricated to as much as 5% of blood agar base, and then incubated at 37 °C for 18–24 h. A positive result indicates that the microbe is pathogenic if a clear zone is formed around the colony on the media [49].

##### DNase Assay

The DNase test was performed using DNase agar (Oxide Ltd., Basingstoke, UK). Bacteria to be tested were inoculated on a DNase agar plate, added with 0.005% methyl green, and then incubated at 37 °C for 24 h. After incubation, the DNase agar plate was immersed in 1 N HCl for 5 min [50]. A positive result, if a clear zone is formed around the colony, indicates that DNase activity hydrolyzes deoxyribonuclease.

##### Gelatinase Assay

The gelatinase test was carried out using bacteria that had been pure-cultured using a loop needle, then inserted into the nutrient gelatin media (Oxide Ltd., Basingstoke, UK) in the middle of the media. Bacteria inoculated into gelatin media were stored in an incubator at 37 °C for ±3 days. After waiting for ±3 days, observe using a test tube containing bacterial isolates stored in the refrigerator at 4 °C. Wait 25 min for the jar in the refrigerator. Furthermore, observations were made, including whether or not the gelatin was melted compared to the control. If there is melting of gelatin, it indicates that the bacteria can produce gelatinase exoenzymes [49].

##### Coagulase Test

This test consists of inoculating a suspension of the bacterial strain in rabbit plasma in a test tube, which was incubated in a bacteriological incubator at 37 °C for 24 h. Over this period, it will be observed whether or not a clot is formed in the plasma, in which the presence of coagulation will indicate a positive result, and the opposite a negative result [50].

##### Elastase Assay

Every 1 mL of elastase was mixed with 20 mg of the Congo red elastin substrate (Sigma-Aldrich, St. Louis, MO, USA) suspended in 2 mL of 0.2 mol/L acid buffer (pH 7.4) with shake (8000× *g*) incubation at 37 °C for 20 min. The reaction was ended by adding 100 μL of 0.12 M EDTA (Sigma-Aldrich, MO, USA) (pH 8.0). Following centrifugation (8000× *g* for 20 min), the absorbance at 495 nm was calculated with a spectrophotometer calibrated to control the elastin-Congo red sample incubated without the enzyme. One unit (U) of elastase was equivalent to the quantity of enzyme, causing a rise in A_495_ by 0.01 after 1 h of incubation under standard assay conditions [51]. 

##### Pyocyanin Assay

Different *P. aeruginosa* strains were cultured in 250 mL conical flasks containing 50 mL of glycerol-supplemented nutrient broth medium and statically incubated at 37 °C; the color of the medium became dark green due to the generation of pyocyanin pigments. For the removal of bacterial cells, the culture medium was subjected to centrifugation at 12,000× *g* for 20 min at 4 °C. The resultant supernatant was filtered through a 0.2 µ syringe filter to obtain clear and green solution of pyocyanin.

The crude pyocyanin pigment previously stored in sterile containers at 4 °C was re-dissolved in 5 mL of chloroform and absorbed into a small quantity of silica gel (mesh size 200–500). Silica-gel-absorbed crude pigment was loaded on a column (30 cm length × 2 cm diameter) equilibrated with 1% methanol in chloroform. Purified pyocyanin was eluted with 15% methanol in chloroform. The eluted fractions were examined by scanning a UV-Vis spectrophotometer (Shimadzu corporation, Kyoto, Japan); fractions with the same λ-max were collected and dried on a rotary evaporator (Sigma-Aldrich, MO, USA) at 37 °C. The purified pyocyanin was subjected to spectroscopic analysis. Ultraviolet and visible absorption spectra of purified pyocyanin dissolved in chloroform or 0.1 N HCl were recorded over 200–700 nm. Mass spectrometric analysis of purified pyocyanin was performed on Shimadzu mass spectrometry GCMS-QP 1000 EX mass spectrometry (Shimadzu corporation, Kyoto, Japan) at 70 eV [52].

##### Fibrinolysin Assay

The bacterial culture was centrifuged (21,000× *g*, 30 min), and the supernatant was discarded. The precipitate was then suspended in 2 mL distilled water and sonicated for 15 s episodes. Following centrifugation at 15,000× *g* for 5 min, the supernatant was harvested, and its fibrinolytic activity was assessed based on plasma plate method. The supernatant (15,000× *g*, 5 min) of 24 h bacterial culture at optimum pH (pH 7) and temperature (40 °C) was harvested and subjected to different treatments. The fibrinolytic activity of treated samples was evaluated based on the plasma plate method. Then, the bacterial culture supernatant was mixed separately with each prepared solution in a 1:1 (*v/v*) ratio and incubated at 40 °C for 60 min. Following treatment, the fibrinolytic activity was evaluated in a plasma plate assay based on the differences in halo zone formation compared to the untreated sample [53].

Gel filtration chromatography was employed to purify fibrinolytic enzymes. The purification factor for clinical sources reached 543.8 U/mL, whereas, for environmental sources, it remained much lower at 250.6 U/mL.

##### Lecithinase Assay

One ml of each bacterial isolate at a cell density of 6 × 10^8^ CFU/mL was inoculated into test tubes containing corn meal broth (Kisan Biotech Ltd., Hyderabad, India) and incubated for 24 h at 37 °C. After incubation, the cultures were centrifuged at 2500 rpm for 15 min to obtain a cell-free filtrate, and 100 μL of the filtrate was transferred into 10 mm wells made centrally in the egg-yolk agar plates and incubated for 24 h at 37 °C. Opaque zones were measured as indicators of lecithinase production, and the means were used as criterion of lecithinase activity [54].

##### Hyaluronidase Assay

One mg/mL of sodium hyaluronate (Sigma-Aldrich, MO, USA) (previously dried in a vacuum oven with phosphorus pentoxide (Sigma-Aldrich, MO, USA) for 48 h) stock solution was diluted before use with an equal volume of 0.02 M phosphate buffer solution (pH 6.3, dissolved 2.5 g of monobasic sodium phosphate (Sigma-Aldrich, MO, USA), 1.0 g of anhydrous dibasic sodium phosphate, and 8.2 g of sodium chloride in water to make 1 L). Then, 10 mg/mL of BSA solution was diluted with 4 volumes of 0.5 M acetate buffer solution (dissolve 14 g of potassium acetate (Sigma-Aldrich, MO, USA) and 20.5 mL of glacial acetic acid in water to make 1 L, and then adjust with 4 M hydrochloric acid to pH 3.1). The standard solution was prepared immediately before use by dissolving standard hyaluronidase in cold hydrolyzed gelatin solution (mix 250 mL of phosphate buffer solution with 250 mL of water and dissolve 330 mg of hydrolyzed gelatin in the mixture) to obtain a solution of 10 U/mL. The sample solution was also properly diluted with cold hydrolyzed gelatin before use. The sample detection reaction temperature was 42 °C [55].

##### Chondroitinase Assay

Chondroitinase (CS) activity was measured according to the UV 232 nm method. The enzymatic reaction was carried out at 37 °C using CS A and CS B as substrates in the same buffer (20 mmol L^−1^ Tris-HCl, pH 7.4). CS A and CS B degradation was monitored by UV absorbance at 232 nm, and the activity was calculated using a molar extinction coefficient of 3800 L mol^−1^ cm^−1^. The protein concentration was detected using the Bradford Protein Assay Kit (Bio-Rad, Dubai, UAE). One international unit was defined as the amount of protein that could form 1 mmol L^−1^ 4,5-unsaturated uronic acid per minute at 37 °C [56].

### 4.5. Inhibitory Potential Evaluation

#### 4.5.1. Extraction of Enzymes from *P. aeruginosa*

The amount of 75 mL of the produced broth containing the bacterial cells, in which the extracellular enzyme was found, was carried into centrifuge tubes, and the bacterial cells were cooled by centrifugation at 10,000× *g* at 4 °C for 10 min. The supernatant, containing extracellular protein, was kept and the pellet containing bacterial cells was discarded [57].

#### 4.5.2. Agar Well Diffusion Assay

Prepare agar plates with suitable growth media and inoculate them with quality strains (*P. aeruginosa* isolates, gram-positive bacteria, *C. albicans*, and gram-negative bacteria). Create wells in the agar and introduce standardized inoculum of each isolated *P. aeruginosa* strain to be tested. After incubation, the inhibitory potential was assessed by measuring the inhibition zones around the wells.

#### 4.5.3. Agar Disk Diffusion Assay

The procedure for conducting the disk diffusion method using isolated *P. aeruginosa* strains from clinical and environmental sources against other pathogenic bacteria and fungi involved several steps. First, the isolated *P. aeruginosa* strains were cultured and grown in MacConkey agar. Pure colony inocula were prepared in physiological saline by adjusting the turbidity of bacterial suspension to 0.5 McFarland standard, which is visually comparable to a microbial suspension of approximately 1.5 × 10^8^ CFU/mL. Optimally, within 15 min after adjusting the turbidity of the inoculum suspension, a sterile cotton swab was dipped into the adjusted suspension. The dried surface of a Mueller–Hinton agar (Oxide Ltd., Basingstoke, UK) plate was inoculated by spreading the swab over the entire sterile agar surface. This procedure is repeated by spreading two more times, rotating the plate approximately 60° each time to ensure an even distribution of inocula. Sterile disks impregnated with purified compounds derived from *P. aeruginosa* strains were placed on the agar surface. These disks contained potential antimicrobial agents. The plates were then incubated at 37 °C for bacterial and fungal growth. After the incubation period, the inhibition zones formed around the disks were measured, indicating the effectiveness of the isolated *P. aeruginosa* strains against the target pathogenic bacteria and fungi. The results obtained from this procedure provide valuable insights into the antimicrobial potential of isolated strains and their suitability for further exploration as potential sources of novel antimicrobial compounds.

## 5. Conclusions

This study presents a comprehensive overview of two investigations into the inhibitory effects of diverse *P. aeruginosa* strains sourced from clinical and environmental origins, targeting both gram-positive and gram-negative bacteria as well as fungi. Through parallel methodologies using disk and well diffusion techniques, coupled with enzyme production assessments via biochemical assays, insights into the antimicrobial potential of *P. aeruginosa* strains were garnered. Notably, significant variations in the enzyme production were noted among clinical strains, with hemolysin and protease prevailing, while gelatinase, chondroitinase, and hyaluronidase exhibited varying frequencies. Conversely, environmental isolates exhibited distinct enzymatic patterns, indicative of adaptation. Inhibition against bacterial strains showcased strain-specificity, particularly against gram-positive bacteria, and limited antifungal effects against *C. albicans*. The study underscores the strain-dependent nature of the inhibitory responses and enzyme secretion, hinting at the potential contributions of specific enzymes like hemolysin and protease in antimicrobial activity. The intricate interplay between enzyme production and pathogen inhibition warrants further exploration. These findings underscore the promising reservoir of *P. aeruginosa* strains for antimicrobial applications, especially targeting gram-positive bacteria, urging future research into the mechanistic underpinnings and therapeutic potentials of these inhibitory phenomena.

## Figures and Tables

**Table 1 antibiotics-12-01354-t001:** The enzyme and virulent factor production by *P. aeruginosa* isolated from clinical sources: a comparative analysis.

	Total [n (%)]	Frequency Among Clinical Isolates (n = 61)			
		Blood [(n = 16) (%)]	Sputum [(n = 15) (%)]	Wound [(n = 20) (%)]	Urine [(n = 10) (%)]
Hemolysin (Virulent factor)	58 (95)	16 (100)	13 (87)	20 (100)	9 (90)
Protease	58 (95)	16 (100)	13 (87)	20 (100)	9 (90)
Gelatinase	53 (87)	12 (75)	13 (87)	20 (100)	8 (80)
Pyocyanin (Pigment and virulent factor)	37 (61)	16 (100)	10 (67)	3 (15)	8(80)
Fibrinolysin (Virulent factor)	46 (75)	12 (75)	8 (53)	20 (100)	6 (60)
Lipase	45 (74)	10 (63)	8 (53)	20 (100)	7 (70)
Coagulase	40 (66)	12 (75)	8 (53)	14 (70)	6 (60)
Lecithinase	37 (61)	6 (38)	5 (33)	20 (100)	6 (60)
Elastase	17 (28)	9 (56)	3 (20)	3 (15)	2 (20)
DNase	8 (13)	3 (19)	2 (13)	0	3 (30)
Chondroitinase	0	0	0	0	0
Hyaluronidase	0	0	0	0	0

**Table 2 antibiotics-12-01354-t002:** The enzyme and virulent factor production by *P. aeruginosa* isolated from environmental sources: a comparative analysis.

	Total [n (%)]	Frequency among Environmental Isolates (n = 30)					
		Locations					
		3/1 [(n = 10) (%)]	3/2 [(n = 6) (%)]	3/3 [(n = 4) (%)]	3/4 [(n = 4) (%)	3/5 [(n = 3) (%)]	22G [(n = 3) (%)]
Hemolysin (Virulent factor)	13 (43.3)	5 (50)	3 (50)	2 (50)	1 (25)	1 (33.3)	1 (33.3)
Protease	25 (83.3)	8 (80)	5 (83.3)	4 (100)	3 (75)	3 (100)	2 (66.7)
Gelatinase	20 (66.7)	6 (60)	4 (66.7)	3 (75)	2 (50)	2 (66.7)	3 (100)
Pyocyanin (Pigment and virulent factor)	0	0	0	0	0	0	0
Fibrinolysin (Virulent factor)	9 (30)	3 (30)	2 (33.3)	1 (25)	1 (25)	1 (33.3)	1 (33.3)
Lipase	18 (60)	6	4 (66.7)	2 (50)	2 (50)	2 (66.7)	2 (66.7)
Coagulase	0	0	0	0	0	0	0
Lecithinase	24 (80)	8 (80)	5 (83.3)	4 (100)	3 (75)	2 (66.7)	2 (66.7)
Elastase	0	0	0	0	0	0	0
DNase	0	0	0	0	0	0	0
Chondroitinase	0	0	0	0	0	0	0
Hyaluronidase	0	0	0	0	0	0	0

**Table 3 antibiotics-12-01354-t003:** Isolation of *P. aeruginosa* strains from clinical specimens: inhibitory effects against gram-positive bacteria and fungus.

Clinical Specimens	Gram-Positive Bacteria						Fungus	
	*S. aureus* ATCC29213		*S. pneumonia* ATCC49619		*B. cereus* ATCC14579		*C. albicans* ATCC10231	
	WD	DD	WD	DD	WD	DD	WD	DD
PA ATCC27853								
24 h	-	-	-	++	+	++	-	++
48 h	+++	++++	++	++	+	++	+++	+++
PA_urine_								
24 h	++	++	-	++	++	++	-	+++
48 h	++	++	+++	++	++	++	+++	+++
PA_wound_								
24 h	-	++	++	++	++	+++	++	++
48 h	-	++	++	++++	+++	+++	++	++
PA_blood_								
24 h	-	-	-	++	++	++	++	++
48 h	-	-	++	++++	++	++	++	++++
PA_sputum_								
24 h	-	++	-	++	-	++	-	++
48 h	++	++	++	++	+	++	+++	+++

WD: Well diffusion method; DD: Disc diffusion method; (-) No inhibition; (+) Weak inhibition; (++) Partial inhibition; (+++) Strong inhibition; (++++) Very strong inhibition; PA_urine_: *P. aeruginosa* isolated from urine; PA_wound_: *P. aeruginosa* isolated from wound; PA_blood_: *P. aeruginosa* isolated from blood; PA_sputum_: *P. aeruginosa* isolated from sputum.

**Table 4 antibiotics-12-01354-t004:** Isolation of *P. aeruginosa* strains from environmental specimens: inhibitory effects against gram-positive bacteria and fungus.

Environmental Sources	Gram-Positive Bacteria						Fungus	
	*S. aureus* ATCC29213		*S. pneumonia* ATCC49619		*B. cereus* ATCC14579		*C. albicans* ATCC10231	
	WD	DD	WD	DD	WD	DD	WD	DD
PA ATCC27853								
24 h	-	-	-	++	+	++	-	++
48 h	+++	++++	++	++	+	++	++	++
PA (3/1)								
24 h	-	++	++	++	-	++	++	++
48 h	-	++	++	++	-	++	++	++
PA (3/2)								
24 h	-	++	++	++	-	++	++	++
48 h	-	++	++	++	-	++	++	++
PA (3/3)								
24 h	-	++	++	++	-	++	++	++
48 h	-	++	++	++	-	++	++	++
PA (3/4)								
24 h	-	+	++	++	-	++	++	++
48 h	-	+	++	++	-	++	++	++
PA (3/5)								
24 h	-	+	+	++	-	++	++	++
48 h	-	+	++	++	-	++	++	++
PA (22G)								
24 h	-	+	++	++	-	++	++	++
48 h	-	+	++	++	-	++	++	++

WD: Well diffusion method; DD: Disk diffusion method; (-) No inhibition; (+) Weak inhibition; (++) Partial inhibition; (+++) Strong inhibition; (++++) Very strong inhibition.

**Table 5 antibiotics-12-01354-t005:** Inhibition degrees of pathogenic gram-positive bacteria and fungi by various *P. aeruginosa* strains: a comparative study using well and disk diffusion methods.

	Well Diffusion Method	Disk Diffusion Method
Control:		
PA ATCC27853	SA & CA > SP > BC	SA.CA > SP & BC
Clinical sources:		
PA_urine_	SP & CA > SA & BC	CA > SA, SP & BC
PA_wound_	BC > SP & CA > SA	SP & BC > SP & CA
PA_blood_	SP, BC & CA > BC	SP & CA > BC > SA
PA_sputum_	CA > SA & SP > BC	CA > SA, SP & BC
Environmental sources:		
PA (3/1)	SP & CA > SA & BC	SA, SP, BC & CA
PA (3/2)	SP & CA > SA & BC	SA, SP, BC & CA
PA (3/3)	SP & CA > SA & BC	SA, SP, BC & CA
PA (3/4)	SP & CA > SA & BC	SP, BC & CA > SA
PA (3/5)	SP & CA > SA & BC	SP, BC & CA > SA
PA (G22)	SP & CA > SA & BC	SP, BC & CA > SA

PA: *Pseudomonas aeruginosa*; SA: *Staphylococcus aureus*; SP: *Streptococcus pneumoniae*; BC: *Bacillus cereus*; CA: *Candida albicans*; PA_urine_: *Pseudomonas aeruginosa* isolated from urine; PA_wound_: *Pseudomonas aeruginosa* isolated from wound; PA_blood_: *Pseudomonas aeruginosa* isolated from blood; PA_sputum_: *Pseudomonas aeruginosa* isolated from sputum.

**Table 6 antibiotics-12-01354-t006:** Isolation of *P. aeruginosa* strains from clinical specimens: inhibitory effects against gram-negative bacteria.

Clinical Specimens	Gram-Negative Bacteria					
	*E. coli* ATCC25922		*K. pneumonia* ATCC700603		*S. enterica* Serotype Typhi ATCC13311	
	WD	DD	WD	DD	WD	DD
PA ATCC27853						
24 h	-	-	+++	-	-	++
48 h	-	-	++++	+	-	++
PA_urine_						
24 h	-	-	++	++	++	+
48 h	-	-	++	++	++	++
PA_wound_						
24 h	-	-	-	++	++	++
48 h	-	++	-	++	++	+++
PA_blood_						
24 h	-	-	-	-	-	-
48 h	-	-	-	-	-	-
PA_sputum_						
24 h	-	-	+++	-	-	++
48 h	-	-	++++	-	-	++

WD: Well diffusion method; DD: Disc diffusion method; (-) No inhibition; (+) Weak inhibition; (++) Partial inhibition; (+++) Strong inhibition; (++++) Very strong inhibition. PA_urine_: *P. aeruginosa* isolated from urine; PA_wound_: *P. aeruginosa* isolated from wound; PA_blood_: *P. aeruginosa* isolated from blood; PA_sputum_: *P. aeruginosa* isolated from sputum.

**Table 7 antibiotics-12-01354-t007:** Isolation of *P. aeruginosa* strains from environmental sources: inhibitory effects against gram-negative bacteria.

Environmental Sources	Gram-Negative Bacteria					
	*E. coli* ATCC25922		*K. pneumonia* ATCC700603		*S. enterica* Serotype Typhi ATCC13311	
	WD	DD	WD	DD	WD	DD
PA ATCC27853						
24 h	-	-	+++	-	-	++
48 h	-	-	++++	+	-	++
PA (3/1)						
24 h	-	++	-	+	++	++
48 h	-	++	-	+	++	+++
PA (3/2)						
24 h	-	++	-	++	++	++
48 h	+	++	-	++	++	+++
PA (3/3)						
24 h	-	++	-	++	++	++
48 h	++	++	-	++	++	+++
PA (3/4)						
24 h	-	+	-	+	++	++
48 h	++	+	-	+	+++	+++
PA (3/5)						
24 h	-	+	-	++	++	++
48 h	-	+	-	++	++	++
PA (22G)						
24 h	-	++	-	++	++	++
48 h	-	++	-	++	++	+++

WD: Well diffusion method; DD: Disk diffusion method; (-) No inhibition; (+) Weak inhibition; (++) Partial inhibition; (+++) Strong inhibition; (++++) Very strong inhibition.

**Table 8 antibiotics-12-01354-t008:** Inhibition degrees of pathogenic gram-negative bacteria by various *P. aeruginosa* strains: a comparative study using well and disk diffusion methods.

	Well Diffusion Method	Disk Diffusion Method
Control:		
PA ATCC27853	KP > EC & ST	ST > KP > EC
Clinical sources:		
PA_urine_	KP & ST > EC	KP & ST > EC
PA_wound_	ST > EC & KP	ST > KP & EC
PA_blood_	No effects	No Effects
PA_sputum_	KP > EC & ST	ST > EC & KP
Environmental sources:		
PA (3/1)	ST > EC & KP	ST > EC > KP
PA (3/2)	ST > EC & KP	ST > EC & KP
PA (3/3)	ST > EC > KP	ST > EC & KP
PA (3/4)	ST > EC > KP	ST > EC & KP
PA (3/5)	ST > EC & KP	ST & KP > EC
PA (G22)	ST > EC & KP	ST > EC & KP

PA: *Pseudomonas aeruginosa*; EC: *E. coli*; KP: *Klebsiella pneumonia*; ST: *Salmonella enterica* serotype Typhi; PA_urine_: *Pseudomonas aeruginosa* isolated from urine; PA_wound_: *Pseudomonas aeruginosa* isolated from wound; PA_blood_: *Pseudomonas aeruginosa* isolated from blood; PA_sputum_: *Pseudomonas aeruginosa* isolated from sputum.

## Data Availability

The datasets used and/or analyzed during the current study is available from the corresponding author upon reasonable request.

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
