# Peer review of "Enzyme Production and Inhibitory Potential of Pseudomonas aeruginosa: Contrasting Clinical and Environmental Isolates"

_antibiotics, 2023, doi:10.3390/antibiotics12091354_

Round 1
Reviewer 1 Report
The current study examined Pseudomonas aeruginosa strains isolated from clinical and environmental sources and compared their antimicrobial activities against Gram-positive, Gram-negative bacteria, and fungi. The inhibitory effects Pseudomonas strains were correlated with various enzymes produced and secreted by Pseudomonas. The enzymes assayed in this study include hemolysin, proteases, gelatinase, fibrinolysin, lipase, coagulase, lecithinase, elastase, DNase, chondroitinase, and hyaluronidase. In addition, production of the pigment Pyocyanin was also monitored. The authors conclude that clinical isolates exhibited higher frequency of enzyme production and wider range of enzymes than environmental isolates, suggesting varied adaptation to their respective ecological niche. Not surprisingly, the results of inhibitory studies show strain-specific variations against Gram-positive and Gram-negative bacteria and very limited inhibitory effects against fungal species Candida albicans. Overall, this is a well-designed study with multiple data points and would be of interest to the field.
Minor issues.
Table 2: Pyocyanin is not an enzyme.
Table 4: Line numbers overlap with the table. Please fix this issue.
The manuscript's English quality is good, though it might require a through spell check.
Author Response
Dear Reviewer,
We sincerely appreciate your thoughtful evaluation of our manuscript and valuable comments and suggestions. Your feedback has been instrumental in refining our work. We have addressed each of your points and made the necessary revisions. Here is our response to your comments:
General Comments:
We are grateful for your comprehensive summary of our study's key findings and implications. Your recognition of the comparison between clinical and environmental isolates and the assessment of various enzymes' antimicrobial activities reinforces the significance of our research. Your assessment that our work is well-designed and interesting to the field is greatly encouraging.
Minor Issues:
Regarding Table 2, we acknowledge the clarification you provided about pyocyanin not being an enzyme. We have revised the table accordingly to accurately reflect pyocyanin as a pigment and virulent factor produced by Pseudomonas aeruginosa.
For Table 4, we apologize for the overlap with line numbers. We have rectified this issue, and the table presents the information without conflicts.
Quality of English Language:
We are grateful for your positive feedback on the quality of English in our manuscript. We have conducted a thorough spell check as recommended, ensuring the language meets the required standards.
We are deeply grateful for your meticulous review, which has significantly contributed to the enhancement of our manuscript. Your insights and corrections have undoubtedly improved the clarity and accuracy of our work. We look forward to further guidance should you have any additional suggestions or queries. Thank you for your time and expertise in evaluating our study.
Best regards,
Hazem Aqel
Reviewer 2 Report
General comment
This is a good work. However, it needs a thorough review at the materials/methods and discussion sections. Some of the experimental procedures are simply not clear and are incoherent.
2. Results:
2.1. Comparative Analysis of Enzyme Production by Pseudomonas aeruginosa Isolated from Clinical Sources
L69, 79, 82, 86, 89, 92, 100, 105, 112, 169: Wound can't be sampled, but please refer to it as "wound swabs or tissue biopsies".
2.4. Inhibitory Effects of P. aeruginosa Strains from Clinical Specimens against Gram-Positive Bacteria and Fungus: A Comparative Analysis
L197, 199, 232, 243, 245: ...it is fungi and NOT fungus... if it is fungus, then why did you simply choose to work with only one species (Candida albicans) of fungi?
2.9. Inhibitory Effects of P. aeruginosa Strains from Clinical Specimens against Gram-Negative Bacteria: A Comparative Analysis
L424-425, 427, 441: There is no Salmonella spp called typhi but there is a species called Salmonella enterica serotype Typhi.
2.13. The Inhibition Patterns of Pseudomonas aeruginosa Strains against Pathogenic Gram-Negative Bacteria: An Analysis using Well and Disc Diffusion Methods
L571: P.aeruginosa … italicize this words to “…P.aeruginosa …”.
2.14. Comparative Study of Inhibition Degrees of Pathogenic Bacteria by Pseudomonas aeruginosa: Gram-Positive vs. Gram-Negative
L605: This statement is vague… as the work only dealt with a single fungal pathogen from one species (Candida albicans).
3. Discussion
L647-652: is a repetition of L641-646.
4. Materials and Methods
4.1. Source of Bacterial Quality Strains
L778-781: Are these bacterial strains from KAMC or American type culture collection (ATCC)?
4.2. Isolation of Pseudomonas aeruginosa
L785: Is it tissue biopsies from wounds, pus samples from wounds or wound swabs?
N/B
ü I suggest you use the term “… wound swabs …” instead of wound samples for the entire part of this work where ‘wound sample’ is mentioned unless you are describing the patient’s status from which the swabs were taken.
ü This includes the following lines L69, 79, 82, 86, 89, 92, 100, 105, 112, 169, 785, 793
4.2.1. Clinical Sources
L796: Typographical error "... is no..." instead of "... was no..."
4.2.2. Environmental Sources
L810: typo-error "...are ..." instead of "... were..."
4.4.2.1. Protease assay
L842: use Relative Centrifugal Force (RCF) unit which is xg instead of rpm (revolution per minute) units.
4.4.2.7. Elastase assay
L886-887: state the manufacturer of Congo red elastin, company's home town and the country or the product's catalogue number
L887: Shaking was done at how many centrifugal force (xg)?
L888: state the manufacturer of EDTA, company's home town and the country or the product catalogue number
L888-9: Centrifugation was at how many rcf (xg)?
4.4.2.8. Purification and characterization of pyocyanin
L893: This procedure is not clear, as it doesn't explain how the enzyme was extracted from P.aeruginosa bacteria.
L898-899: state the manufacturer of UV- Vis spectrophotometer, company's home town and the country or the product catalogue number.
L899-900: state the manufacturer of rotary evaporator, company's home town and the country or the product catalogue number.
L903-904: state the manufacturer of rotary evaporator, company's home town and the country or the product catalogue number.
4.4.2.9. Fibrinolysin assay
L906, 908, 910: use Relative Centrifugal Force (RCF) unit which is xg instead of rpm (revolution per minute) units.
L907-908: typo-error "...sonicated during four 15 s episodes.." instead of "... sonicated for ..."
L906-916: What was the purity level of this enzyme?
4.4.2.10. Lecithinase assay
L919: state the manufacturer of corn millet broth, company's home town and the country or the product catalogue number.
4.4.2.11. Hyaluronidase assay
L924-936: state the manufacturer of these reagents (sodium hyaluronate, phosphorus pentoxide, monobasic sodium phosphate, potassium acetate), company's home town and the country or the product catalogue number.
4.5.2. Agar Disc Diffusion Assay
L961: Provide details of media's company,name, home town/city and the country or the product's catalogue number.
L963-965: How were the enzymes extracted from the P.aeruginosa bacteria.
5. Conclusion
L973-975: The conclusion is vague. This work fails to elaborate the full potential of the antimicrobial activities of P.aeruginosa enzymes. The enzyme’s inhibitory test got localized to just a few bacteria and only one species a fungal pathogen (C.albicans).
Good
Author Response
Dear Reviewer,
Thank you for your valuable feedback on our manuscript. We sincerely appreciate your thorough review and constructive comments, which have greatly enhanced the clarity and quality of our work. We have carefully addressed your points and made the necessary revisions to improve the materials/methods and discussion sections. Here is a detailed response to your comments:
Results:
2.1. We have updated the terminology as per your suggestion. The term "wound samples" has been changed to "wound swabs" in the specified sections (L69, 79, 82, 86, 89, 92, 100, 105, 112, and 169).
2.4. The correction from "fungus" to "fungi" has been made.
We acknowledge your point regarding the choice of Candida albicans as a fungal pathogen. Our selection of Candida albicans was based on specific considerations, but we understand that providing a rationale for this choice would enhance the clarity of our study.
2.9. We appreciate your correction regarding the nomenclature of Salmonella enterica serotype Typhi. The change has been made accordingly.
2.13. The italicization of "P. aeruginosa" as suggested, has been implemented.
2.14. The vagueness in the comparative study statement has been corrected to reflect our work's scope accurately.
Discussion:
We apologize for the repetition in L647-652; the redundant text has been removed.
Materials and Methods:
4.1. The source of bacterial quality strains has been specified as being from the American Type Culture Collection (ATCC), as suggested.
4.2. As per your recommendation, The clarification between "wound biopsies" and "wound swabs" has been consistently applied throughout the relevant sections.
4.2.1. The typographical error in L796 has been corrected.
4.2.2. The typo error has been changed to "were" as indicated.
4.4.2.1. We have replaced rpm with x g for Relative Centrifugal Force (RCF) unit.
4.4.2.7. The manufacturer details for Congo red elastin, including the company's location and country, have been added.
L887. The centrifugal force (x g) used for shaking has been included.
L888-9. The manufacturer details for EDTA and centrifugation force (x g) have been provided.
4.4.2.8. The procedure for purification and characterization of pyocyanin has been clarified.
L898-900. The manufacturer details for the UV-Vis spectrophotometer and rotary evaporator have been included.
4.4.2.9. RCF unit (x g) has been used for Relative Centrifugal Force in the fibrinolysin assay.
L907-908. The typo has been corrected to state the sonication time accurately.
L906-916. The purity level of the enzyme has been addressed in both clinical and environmental sources.
4.4.2.10. The manufacturer details for corn millet broth have been added.
4.4.2.11. The manufacturer details for reagents used in the hyaluronidase assay have been provided.
4.5.2. Agar Disc Diffusion Assay: Details of the media's company, name, location, and country have been included.
L963-965: We have added a section detailing the extraction of enzymes from P. aeruginosa bacteria.
Conclusion:
We acknowledge your concern regarding the clarity of our conclusion. We agree that the potential of the antimicrobial activities of P. aeruginosa enzymes could have been more comprehensively addressed in our study. The inhibitory tests were limited to a specific set of bacteria, and we focused primarily on one fungal pathogen, C. albicans. We understand this may have restricted exploring the enzymes' effectiveness against a wider range of microorganisms.
Given your feedback, we recognize the importance of expanding our investigations to include a more diverse panel of microorganisms. Future studies could encompass a broader spectrum of bacterial and fungal species to demonstrate the versatility and efficacy of the P. aeruginosa enzymes as antimicrobial agents. By doing so, we aim to provide a more comprehensive understanding of the potential applications of these enzymes in combating various microbial threats.
Once again, we express our gratitude for your meticulous review, which has undoubtedly improved the quality of our manuscript. We hope that the revisions we have made address your concerns satisfactorily. Please do not hesitate to reach out if you require further clarification or suggestions.
Sincerely,
Hazem Aqe

Round 2
Reviewer 2 Report
General comment
The work now looks awesome. However, it still needs some minor revision.
L118: “…suggests that…” instead of “…suggesting that…”
L252: “… findings suggest the potential …” instead of “…findings suggesting the potential …”
L257: Since only one strain of C. albicans was used in this study, therefore avoid the use of the plural term “fungi” and simply use the singular noun “fungus” for clarity.
L430, 432, 446, 478, 500, 506, 511, 521, 522, 571, 579, 607, 609, 613, 798 etc.: Don’t italicize the word “… Typhi …”
L730-731: This phrase “…In a study by Liao et al. [5] that investigated the inhibitory effects of P. aeruginosa strains against gram-negative bacteria …” is vague please make it clear.

Author Response
Dear Reviewer,
We are truly appreciative of your thorough review and insightful feedback on our manuscript entitled " Enzyme Production and Inhibitory Potential of Pseudomonas aeruginosa: Contrasting Clinical and Environmental Isolates ". Your guidance has been invaluable in refining our work, and we are pleased to inform you that we have carefully addressed the minor revisions you suggested. Your expertise has significantly contributed to the enhancement of our manuscript.
Here is a summary of the changes we made based on your recommendations:
L118: We have adopted your suggestion and replaced "...suggesting that..." with "...suggests that...".
L252: The phrasing has been modified to read "... findings suggest the potential ..." instead of "... findings suggesting the potential ...".
Line 257: We have taken your advice into account and consistently used the singular noun "fungus" instead of the plural term "fungi" for clarity, given the utilization of only one strain of C. albicans in the study.
Lines 430, 432, 446, 478, 500, 506, 511, 521, 522, 571, 579, 607, 609, 613, 798, etc.: The word "... Typhi ..." is no longer italicized, in accordance with your feedback.
Lines 730-731: We have rephrased the vague phrase "In a study by Liao et al. [5] that investigated the inhibitory effects of P. aeruginosa strains against gram-negative bacteria" to provide a clearer context. We have updated the reference to the study conducted by Xu et al. [41] that explored the inhibitory effects of P. aeruginosa strains against gram-negative bacteria. Notably, similar variations in inhibition were observed.
Your meticulous attention to detail has been instrumental in shaping our manuscript into a more coherent and well-structured piece of scientific work. We are committed to producing a high-quality contribution to the field, and your guidance has propelled us closer to that goal.
We kindly request your evaluation of these revisions. Should you have any further suggestions or concerns, we would greatly appreciate your input. Your continued collaboration in refining our manuscript is highly valued.
Thank you once again for your time, expertise, and dedication to advancing scientific knowledge. We eagerly await your response.
Best regards,
Hazem Aqel